

# A review of deep learning methods in aquatic animal husbandry

Marzuraikah Mohd Stofa, Fatimah Az Zahra Azizan and
Mohd Asyraf Zulkifley

Department of Electrical, Electronic and Systems Engineering, Universiti Kebangsaan Malaysia,
Bangi, Selangor, Malaysia

## ABSTRACT

Aquatic animal husbandry is crucial for global food security and supports millions of livelihoods around the world. With the growing demand for seafood, this industry has become economically significant for many regions, contributing to local and global economies. However, as the industry grows, it faces various major challenges that are not encountered in small-scale setups. Traditional methods for classifying, detecting, and monitoring aquatic animals are often time-consuming, labor-intensive, and prone to inaccuracies. The labor-intensive nature of these operations has led many aquaculture operators to move towards automation systems. Yet, for an automation system to be effectively deployed, it needs an intelligent decision-making system, which is where deep learning techniques come into play. In this article, an extensive methodological review of machine learning methods, primarily the deep learning methods used in aquatic animal husbandry are concisely summarized. This article focuses on the use of deep learning in three key areas: classification, localization, and segmentation. Generally, classification techniques are vital in distinguishing between different species of aquatic organisms, while localization methods are used to identify the respective animal's position within a video or an image. Segmentation techniques, on the other hand, enable the precise delineation of organism boundaries, which is essential information in accurate monitoring systems. Among these key areas, segmentation techniques, particularly through the U-Net model, have shown the best results, even achieving a high segmentation performance of 94.44%. This article also highlights the potential of deep learning to enhance the precision, productivity, and sustainability of automated operations in aquatic animal husbandry. Looking ahead, deep learning offers huge potential to transform the aquaculture industry in terms of cost and operations. Future research should focus on refining existing models to better address real-world challenges such as sensor input quality and multi-modal data across various environments for better automation in the aquaculture industry.

Corresponding author
Mohd Asyraf Zulkifley,
asyraf.zulkifley@ukm.edu.my

## INTRODUCTION

Millions of people around the world depend on aquatic animals for their livelihoods, thus making it crucial to preserve this global food source. Aquatic animal husbandry

encompasses the activities of raising diverse species such as fishes, shrimps, crabs, scallops, corals, jellyfish, aquatic macroinvertebrates, and phytoplanktons (*Sun, Yang & Xie, 2020*). This industry has experienced rapid growth in recent decades, driven by the increasing global demand for seafood. However, the rising demand has led to the depletion of aquatic animal populations and various other environmental issues (*Zhao et al., 2021*). To mitigate some of these issues, deep learning methods have been proposed for aquatic animal husbandry through various automation processes (*Saleh et al., 2024*). Deep learning, a subset of artificial neural networks, has gained a lot of attention recently in the machine learning community because of its deep network layers capability, instead of shallow architecture to produce complex learning representations of data that are more accurate (*Stofa, Zulkifley & Mohamed, 2024*). One of the driving factors of deep learning's state-of-the-art performance can be attributed to the utilization of convolutional neural networks (CNNs), which allow the machines to learn unique patterns from large datasets with high accuracy, that have transformed many automation tasks, especially image classification and object recognition (*Akhyar et al., 2024*).

Classifying or detecting aquatic animals using manual methods is time-consuming and requires high sampling efforts. On the other hand, a deep learning-based technology is capable of automating these complex tasks by using accurate classification and detection techniques without human intervention (*Maluazi, Zulkifley & Kadim, 2024*). Hence, this artificial intelligence (AI) technology, in particular, deep learning has emerged as a promising solution to tackle the issues faced by the aquatic animal husbandry industry. In general, deep learning models possess the remarkable capability to perform automation of complex tasks like image classification, localization, and segmentation with high accuracy, as they can automatically extract relevant features automatically from datasets (*Vo et al., 2021*). Furthermore, as Industry 4.0 has grown, large numbers of AI-driven sensors are used to track the state and performance of animal husbandry operations, offering real-time information about farm conditions (*Maluazi, Zulkifley & Kadim, 2024*). These advanced capabilities have spearheaded the development of numerous deep learning applications in the aquatic animal husbandry domain. Implementing these deep learning technologies can significantly improve the operation efficiency of the farms, increasing their overall productivity and enhancing the sustainability of the industry.

Compared to conventional techniques, deep learning models regularly produce higher accuracy and have demonstrated top performance in various applications, including detecting underwater aquatic animal images. While public datasets such as those on Kaggle provide valuable training resources, the images may not always be clear or high-quality enough due to unique underwater challenges like poor lighting. However, they still offer substantial utility for training a good automation model for aquaculture industry usage.

Unlike previous articles that primarily focus on underwater object detection (*Er et al., 2023*; *Xu et al., 2023*), our study focusses on the role of deep learning in classification, localization, and segmentation tasks. Additionally, we also address issues related to dataset quality, model generalization, and real-time deployment, which are not the main focus of existing reviews. This review aims to provide insights for both AI researchers, who focus on improving model architectures, and aquaculture practitioners,

who seek practical AI-driven solutions for real-world challenges. By bridging these perspectives, this review article captured the main important points from research in recent years that has focused on detecting underwater images using deep learning methods. The general framework of the deep learning-based system includes pre-processing, feature extraction, classification, localization, and segmentation phases, which are the usual steps used in designing accurate automation systems for identifying the species of aquatic organisms in underwater images. These steps ensure that the deep learning model can effectively differentiate between various species, leading to better management of the aquaculture farms. The primary research question guiding this review are: (1) How have deep learning techniques contributed to the automation of monitoring processes in aquatic animal husbandry? (2) What are the strengths and weaknesses of different deep learning techniques (classification, localization, segmentation) used in this field? (3) What are the current challenges and future directions for improving deep learning-based automation in aquaculture? These questions are addressed systematically through the review of recent studies and comparative analyses of deep learning model, Furthermore, this article also reviews and compares key tasks in automated aquaculture farms, which are classification, localization, and segmentation techniques in classification, localization, and segmentation section. Additionally, a comparative analysis using deep learning methods for classification, localization, and segmentation of aquatic animal husbandry was performed in general discussion, highlighting the strengths and weaknesses of each method, followed by a conclusion and future works section.

## SEARCH METHODOLOGY

This literature review aimed to provide a comprehensive understanding of deep learning applications in automating processes within aquatic animal husbandry, specifically focusing on the critical tasks of classification, localization, and segmentation. Given the increasing importance of precision in monitoring aquatic species, the review sought to examine studies that demonstrate how deep learning models can support more efficient and accurate automation within this field. To identify relevant literature, an extensive search was carried out in major academic databases, including IEEE Xplore, Google Scholar, and ScienceDirect (Elsevier). These platforms were selected for their extensive collections of high-quality, peer-reviewed research across engineering, computer science, and applied sciences. The search was conducted using combinations of the following keywords: "Automated" AND "Aquatic" AND "Deep Learning", "Underwater Object Detection" AND ("YOLO" OR "Faster R-CNN" OR "EfficientDet"), "Semantic Segmentation" AND "Underwater Images", and "Real-time Object Detection" AND "Aquaculture".

The inclusion criteria for selecting studies in this review were carefully defined to ensure relevance, quality, and alignment with the objectives of aquatic automation research. First, only studies published between 2018 and 2024 were considered, ensuring the incorporation of recent developments in deep learning and its applications to aquaculture. Second, selected studies had to apply deep learning models directly related to aquatic animal monitoring encompassing species recognition, behavior tracking, population
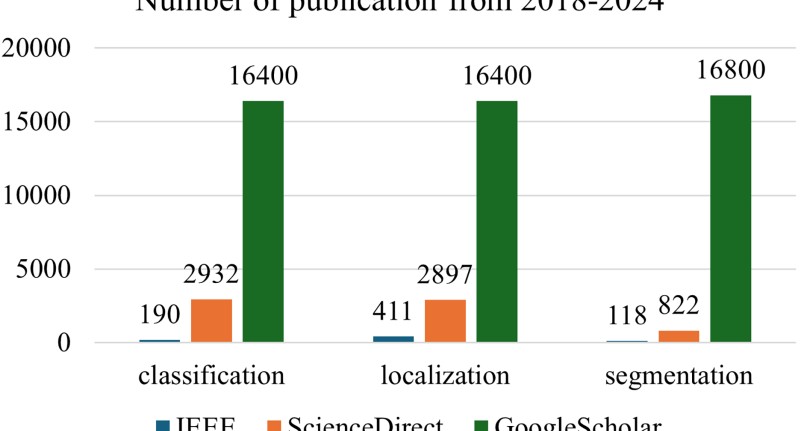

**Figure 1 Publication statistics on aquatic related topics from 2018 to 2024.**

counting, and environmental sensing within aquatic environments. Third, the review focused on articles that addressed at least one of the three primary tasks critical to automation in aquaculture: classification, localization, and segmentation of aquatic enviromenta. These tasks form the foundation for intelligent decision-making in aquaculture systems, enabling enhanced monitoring precision and operational efficiency. Studies that included comparative analyses of different deep learning architectures, reported performance metrics, or demonstrated real-world deployment scenarios were given higher priority in the selection process.

Figure 1 shows the number of publications on aquatic related topics from 2018 to 2024. The exclusion criteria were defined to maintain the focus and relevance of this review within the scope of deep learning applications for aquatic animal husbandry. Studies were excluded if they focused solely on traditional machine learning approaches without incorporating or comparing deep learning methods, as the aim of this review is to evaluate advancements specific to deep learning technologies. Additionally, research centered on non-aquatic animal domains was excluded to ensure thematic consistency with the aquatic focus of this work. Finally, articles lacking experimental validation, including those that provided only conceptual discussions or unverified model proposals, were omitted to ensure that the analysis was grounded in empirical evidence and reproducible results. This ensured that all included studies contributed meaningfully to understanding the current capabilities, limitations, and opportunities of deep learning models in aquaculture automation.

Approximately 150 articles were initially retrieved. After applying the inclusion/exclusion criteria and reviewing abstracts and full texts, 52 articles were selected for detailed analysis. Studies were evaluated based on relevance, citation impact, methodological rigor, and the clarity of results reported. These databases provided access to a diverse range of sources, allowing for a robust analysis that captures both established and emerging applications of deep learning within aquatic animal husbandry.

**Table 1 Applications of deep learning methods in aquatic animal husbandry: classification techniques.**

| Method | Datasets | Aquatic animal classes | Deep learning techniques | Performance measures | Strength | Weakness |
|---|---|---|---|---|---|---|
| Saleh, Laradji & Konovalov (2020) | DeepFish | 20 different habitats | ResNet-50 | Accuracy: 0.99 | Large dataset providing a diverse range of information for model training | The complexity of underwater scenes makes it challenging for models to perform |
| Shammi et al. (2021) | Fish-Pak | Six different species of fishes | CNN | Accuracy: 88.96% | CNN resulted in a high classification accuracy, which is significantly better than traditional algorithm | Limited scope, only six types of fish were classified |
| Pagire & Phadke (2022) | Kaggle | Nine different breeds of fishes | MobileNet | Accuracy: 99.74% | High accuracy in detecting and classifying nine different fish breeds | Underwater environment challenges, such as varying light conditions that can affect detection accuracy |
| Mol & Jose (2022) | Kaggle | Nine fish species | AlexNet GoogleNet VGGNet | Accuracy; AlexNet: 94% GoogleNet: 95% VGGNet: 95.5% | Use a modified version of the AlexNet model that has three fully connected layers and five convolutional layers | Lack of detailed information on dataset size and diversity |
| Bhanumathi et al. (2022) | Kaggle | Three fish species | AlexNet | Accuracy: 90% | Achieves accuracy higher than the standard CNN model | The model's performance may be constrained by the dataset size |
| Ben Tamou, Benzinou & Nasreddine (2022) | LiteClef 2015 | 15 fish species | ResNet50 | Accuracy: 81.83% | The use of underwater video systems allows for the study of marine biodiversity without disturbing the environment | Poor quality images, which can complicate the automatic analysis process |
| Paraschiv et al. (2022) | Symbiosis-NCC QUT-NOA | Six fish species | VGG16 | Accuracy: Almost 42% (picture) Almost 49% (from video) | These models use fewer computational resources, making them suitable for long-term, unattended operations | The accuracy of small models is relatively low, around 42% for six fish species, which might not be adequate for all needs |
| Dey et al. (2023) | Author's private datasets | Nine species of marine fauna | MobileNetV2 | Accuracy: 99.83% | The optimized model achieves a 99.83% accuracy, indicating its effectiveness in classifying marine species | Complex underwater pose challenges for classification |

(Continued)

| Method | Datasets | Aquatic animal classes | Deep learning techniques | Performance measures | Strength | Weakness |
|---|---|---|---|---|---|---|
| *Tiwari et al. (2023)* | Marine Animal | Fish, Goldfish, Harbor seal, Jellyfish, Lobster, Oyster, Sea turtle, Squid, Starfish | Inception ResNetV ResNet50 VGG16 InceptionV3 Xception DenseNet MobileNet NASNet | The best validation performance; Inception: 96.75 ResNetV: 93.75 ResNet50: 92.75 MobileNet: 92.73 NASNet: 94.75 | Models were fine-tuned using transfer learning, which allows them to leverage pre-trained knowledge from ImageNet, enhancing their performance on the marine animal dataset | Models perform well on the specific dataset, their ability to generalize to new, unseen data or different marine environments may be limited |
| *Ishwarya et al. (2024)* | Sathyabama Institute of Science and Technology | 20 different categories of underwater marine organisms | AlexNet DarkNet19 Squeeze Net | Accuracy: AlexNet: 70% DarkNet19: 96% Squeeze Net: 98% | Higher accuracy in classifying underwater marine species | Require significant computational resources, including powerful GPUs and extensive training time |

Various deep learning architectures have been applied in this domain, include CNNs, which excel in feature extraction and image recognition. Region-based CNNs (*e.g.*, Faster R-CNN) enhance object detection by refining bounding box predictions, while single-stage detectors like You-Only-Look-Once (YOLO) provide real-time detection capabilities. A comparative analysis of these architectures, including their accuracy and computational efficiency, is provided in Tables 1–3.

This review focuses on three main automation tasks, which are classification, localization, and segmentation that are frequently used in automated aquatic animal husbandry systems. Classification, localization, and segmentation are fundamental tasks in deep learning applications for aquatic animal husbandry. These tasks enable the automated identification of species, precise tracking of individual organisms, and accurate delineation of object boundaries in underwater environments. A classification task is critical for identifying different aquatic animal species, which helps ensure biodiversity is accurately monitored. A localization task allows us to pinpoint the exact location of animals in their environment, which is crucial for understanding their behavior and habitat. A segmentation task provides a detailed segmentation map of the fish boundaries, enabling precise assessments of their health and conditions. These three tasks were selected because they form the backbone of many deep learning applications in this field, each addressing specific needs that are vital for improving automation and decision-making in aquaculture. Given the challenges of murky water conditions, varying lighting, and species diversity, these three tasks were selected as they have the highest impact on improving aquaculture monitoring systems. Figure 2 shows how classification, localization, and segmentation tasks are used in the context of aquatic animal husbandry. The search phrases contain the following terms to reflect the intended goal of the article, which are "Automated" AND "Aquatic" AND "Deep Learning". The resultant articles are then

**Table 2 Applications of deep learning methods in aquatic animal husbandry: localization techniques.**

| Methods | Datasets | Aquatic animal classes | Deep learning techniques | Performance measures | Strength | Weakness |
|---|---|---|---|---|---|---|
| *Xu et al. (2020)* | Author's private datasets | Bluefin tuna | RCNN | Detection rate: 91.5%<br>Accuracy: 92.4% | High accuracy of 92.4% and detection rate of 91.5% for bluefin tuna | Requires a powerful GPU for efficient training |
| *Hu et al. (2020)* | National Natural Science Foundation of Chine Underwater Robot Competition | Sea urchins | SSD | AP:81.0% | Algorithm improves detection accuracy by 7.6% over the classic SSD | The added feature enhancement and cross-level fusion increase the model's complexity |
| *Mathias et al. (2022)* | LCF-15<br>UWA<br>Bubble vision<br>DeepFish | Fish | YOLOv3 | Accuracy:<br>98.5%<br>96.77%<br>97.99%<br>95.3% | Achieves 95.3% to 98.5% accuracy in fish identification across different datasets | Struggles with detecting camouflaged objects in the background |
| *Li et al. (2022)* | URPC-2019 | Starfish, Echinus, Scallop, Holothurian | SSD | Accuracy: 79.76%<br>FPS: 18.95 fps | It operates at 18.9 FPS, 2.8 FPS faster than the original SSD, making it suitable for real-time use | The use of ResNeXt-50 and attention mechanisms can make the model more complex and resource-intensive to implement |
| *Sangari et al. (2023)* | Author's private datasets | Marine organisms | RCNN | Accuracy: 97.89 | Uses RCNN to improve detection in poorly lit underwater images, increasing accuracy | Integrating RCNN and CFTA can be complex, requiring more resources and expertise |
| *Wei et al. (2023)* | RUIE | Sea urchins, Sea cucumbers, Scallops, Starfish | YOLOv5s | AP: 84.32% | Faster processing and efficiency | N/A |
| *Liu et al. (2023a)* | Label fishes in the wild | Rockfish | YOLOv7 | mAP: 94.4%<br>Precision: 99.1%<br>Recall: 99% | With a mAP of 94.4%, the enhanced model outperforms the original YOLOv7 model by 3.5% | Limit its generalizability to other datasets or environments |
| *Raavi, Chandu & SudalaiMuthu (2023)* | Author's private datasets | Silver moony, Bluefin trevally, Puffer fish, Box fish and *etc.* | YOLOv3<br>SSD | Precision: 0.88<br>Recall: 0.86<br>IoU: 0.75 | Improve accuracy in detecting multiple underwater objects | Struggles with detecting objects in complex underwater environments |
| *Rasool, Annamalai & Natarajan (2024)* | Github repository Google images | Dolphin, Fish, Turtle, Jellyfish, Starfish, Swordfish | YOLOv8<br>Faster R-CNN | YOLOv8; mAP: 96.1<br>Faster R-CNN; mAP: 96.4 | Uses advanced deep learning models like YOLOv8 and Faster R-CNN to improve detection speed and accuracy | Underwater settings have complex backdrops and poor image quality |
| *Jain (2024)* | Brackish | Crab, Fish-big, Fish-school, Fish-small, Shrimp, Jellyfish | EfficientDet | IoU: 88.54% | The modified EfficientDet model achieves high accuracy, demonstrating its effectiveness in challenging underwater environments | The study is based on a specific dataset (Brackish-Dataset), which may limit the generalizability of the findings to other underwater environments or datasets with different characteristics |

**Table 3 Applications of deep learning methods in aquatic animal husbandry: segmentation techniques.**

| Methods | Datasets | Aquatic animal classes | Deep learning techniques | Performance measures | Strength | Weakness |
|---|---|---|---|---|---|---|
| *O'Byrne et al. (2018)* | SYNTHIA | Background, Soft fouling, Structure | SegNet | mIoU: 87%<br>Mean accuracy: 94% | Trains models with synthetic images, solving the problem of limited real underwater datasets | Synthetic data might not fully capture real-world underwater complexities |
| *Liu & Fang (2020)* | Public resources on the Internet<br>Video images taken by a laboratory underwater robot (HUBOS-2K, Hokkaido University) | Nautilus, Squid, Plant, Coral, Fish, Jelly fish, Dolphin, Sea lion, Syngnathus, Turtle, Starfish, Shrimp, Octopus, Seahorse, Person, Stone | DeepLabv3+ | mIoU: 64.65% | Improves segmentation accuracy by 3% over the original method | The addition of modules and layers may increase the complexity of the model, potentially requiring more computational resources and time for training |
| *Islam et al. (2020)* | SUIM | Fish, Reefs, Aquatic plants, Wrecks/Ruins, Human divers, Robots, Sea-floor | SUIM-Net | Params: 3.864 M<br>Frame Rate: 28.65 fps<br>mIoU: 77.77 ± 1.64<br>F1-score: 78.86 ± 1.79 | SUIM-Net is designed for fast and efficient processing, suitable for real-time applications | May not perform well in different underwater conditions not covered by the dataset |
| *Nezla, Haridas & Supriya (2021)* | Fish4 Knowledge | Species 08 *Acanthurus nigrofuscus* | U-Net | IoU: 0.8583 | Attained an average IoU score of 0.8583 for a particular class of fish, demonstrating efficient segmentation | Performance is validated only on the Fish4Knowledge dataset |
| *Drews-Jr et al. (2021)* | NAUTEC UWI | Foreground, Background | SegNet DeepLabv3+ | DeepLab;<br>mIoU: 0.919<br>Segnet;<br>mIoU: 0.825 | Employs advanced deep learning models for better segmentation results in underwater images | The real underwater image dataset is small, which may limit model performance |
| *Zhang, Gruen & Li (2022)* | Author's private datasets | Pocillopora corals | U-Net | mIoU: 93.6% | The method effectively distinguishes between living and dead corals, which is important for assessing coral health | Coral boundaries are irregular, making segmentation challenging with traditional metrics |
| *Lin, Tseng & Li (2022)* | Author's private datasets | Coralfish, Background | SUR-Net | F1-score: 95.04%<br>mIoU: 88.19% | Achieves 95.04% F1-score and 88.19% mIoU, showing strong results in detecting and segmenting fish | Use many layers and blocks that can make the model complex and resource-intensive |
| *Han et al. (2023)* | DeepFish | Fish, Background | IST-PSPNet | mIoU: 91.56%<br>Params: 46.68M<br>GFLOPS: 40.27 G | The network improves segmentation accuracy for fish with similar colours and backgrounds | The method is mainly tested on the DeepFish dataset only which might not cover all underwater conditions |
| *Chicchon et al. (2023)* | SUIM<br>RockFish<br>DeepFish | Water background, Sea-floor/obstacles, Fish | U-Net DeepLabv3+ | U-Net;<br>mIoU: 86.10<br>mHD95: 26.53<br>DeepLabv3+;<br>mIoU: 84.85<br>mHD95: 27.65 | Combines active contour theory and level set methods to enhance spatial detail in segmentation | Existing methods often produce low-resolution results, which lack detail for effective monitoring |
| *Zhang, Li & Seet (2024)* | Fish4 Knowledge | Fish, Background | U-Net | mIoU: 94.44%<br>mPA: 97.03%<br>Frame rate: 43.62 fps | Achieves 94.44% mIoU and 97.03% mPA, ensuring precise segmentation | Used of multiple modules can make the model complex to implement |

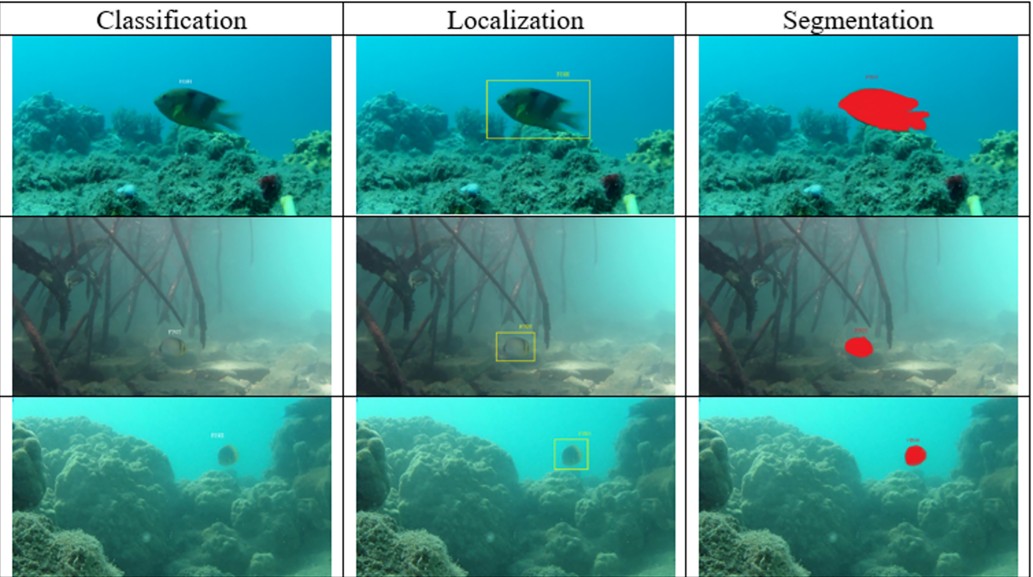

**Figure 2 An overview of classification, localization and segmentation tasks of deep learning methods used for aquatic animal husbandry.**

filtered to focus only on three majority subtasks, which are classification, localization, and segmentation. Some of the subtasks such as regression, clustering, and tracking are excluded from this review due to the limited number of articles in the respective subfields. Furthermore, articles that discuss both the deep learning and conventional machine learning approaches are also accepted since the deep learning approach always results in better automation performance, which is in line with the goal of this article.

## CLASSIFICATION

In the aquaculture industry, some of the important automation tasks are to identify and classify aquatic animal species. Many researchers have shown the potential of deep learning models to classify aquatic species based on underwater images. Among the deep learning-based models, CNNs have emerged as the dominant architecture used for image classification tasks, exhibiting superior performance compared to conventional machine learning algorithms (*Sun, Yang & Xie, 2020*). It is also interesting to note that the deep learning approach, specifically models that are based on the CNN classifier performs better than the average human in many tasks involving object classification (*Stofa, Zulkifley & Zainuri, 2021*). CNNs can automatically extract hierarchical features from the input data, which makes them particularly well-suited for processing and analyzing visual data. This advanced network capability allows the model to develop highly effective discriminative characteristics for differentiating among various aquatic animal species.

*Bhanumathi & Arthi (2022)* underlines the vital necessity of creating automated fish species classification systems to facilitate biological research in disciplines such as ecology and genetics, where precise species identification is crucial. It examines the shortcomings of existing computer vision methodologies in underwater settings, which frequently

experience indistinct textural characteristics and poor classification efficacy. The article offers a thorough summary of current developments, encompassing algorithms, datasets, performance measurements, and tools utilized in fish identification. *Saleh, Sheaves & Rahimi Azghadi (2022)* presents a comprehensive review of computer vision techniques and deep learning concepts that are relevant to underwater image analysis, with a particular emphasis on the obstacles presented by the poor image quality that is distinctive to aquatic environments. By conducting a review of recent publications, the authors emphasize the necessity of more comprehensive fish monitoring solutions and the existence of research gaps. The fundamentals of CNN architecture are also explained in the article to facilitate comprehension of their potential.

Classifying underwater aquatic animal species presents a significant challenge as highlighted by *Pagire & Phadke (2022)*. To address this challenge, they introduced a MobileNet model designed to detect and recognize nine different breeds of fish using a Kaggle dataset, which is publicly available online. Before implementing the model, the dataset underwent preprocessing steps to optimize the quality of the input data. As a result, the model achieved an impressive accuracy of 99.74%. In a similar study (*Dey et al., 2023*), the authors also tackled the classification problem of nine species of marine fauna, achieving an even higher accuracy of 99.83%, surpassing the previous result in *Pagire & Phadke (2022)* by a factor of 0.09%. Both studies demonstrated nearly flawless classification performance, underscoring the efficiency of deep learning models in this domain.

In addition to these studies, other researchers have explored various approaches to increase the precision and efficiency of aquatic species classification. For instance, a popular deep CNN model, AlexNet was employed in *Mol & Jose (2022)* to classify fish species, also using a Kaggle-based dataset. This model utilizes five convolutional layers to effectively extract texture and color features, complemented by three fully connected layers for feature selection and classification. The performance of this approach was validated through comparative analysis with other leading deep learning models, including GoogleNet and VGGNet. The study reported classification accuracies of 94%, 95%, and 95.5% for AlexNet, GoogleNet, and VGGNet, respectively. In another study, AlexNet was applied to a different Kaggle dataset containing a lesser number of species of only three types, while also achieving a lower accuracy of 90% (*Bhanumathi et al., 2022*).

*Oion et al. (2023)* introduce a deep learning framework based on CNNs for the automated classification of marine species. This framework addresses the limitations of traditional methods and manual identification. It delineates a comprehensive methodology that encompasses data collection, preprocessing, CNN model design, and supervised training with validation to prevent overfitting. The system demonstrated outstanding results in metrics, including precision, recall, accuracy, and F1-score, when assessed against a dataset that included 23 marine species classes. The final model obtained an accuracy of 87.99%, surpassing previous methods, and interpretability was improved through the use of techniques such as Grad-CAM. Then, *Zheng et al. (2022)* propose KRS-Net, a deep learning classification network aimed at accurately identifying 13 koi fish varieties that demonstrate excellent visual similarity. KRS-Net, constructed on the AlexNet framework, utilizes residual blocks and contains a support vector machine (SVM) to

enhance feature extraction and classification precision. A dataset including 569 koi photos was augmented to yield 1,464 images, resulting in a model test accuracy of 97.90%. The comparative investigation revealed that KRS-Net surpasses established models such as AlexNet and VGG16, exhibiting structural benefits including improved convergence, less complexity, and enhanced training efficiency.

The dataset used in *Saleh, Laradji & Konovalov (2020)* includes almost 40,000 underwater images taken from 20 tropical marine ecosystems in tropical Australia, sourced from the DeepFish dataset. This diverse set of images captures a range of complex underwater scenes. The study utilized ResNet-50, achieving a notable accuracy of 0.99. In contrast, *Ben Tamou, Benzinou & Nasreddine (2022)* applied a similar technique to the LifeClef 2015 Fish (LCF-15) dataset, comprising over 22,000 annotated images and 700,000 video frames, demonstrated the model's robustness under poor lighting and noisy conditions. The incremental learning strategy outperformed non-incremental methods, achieving a final accuracy of 81.83% and maintaining high precision for difficult species. *Albin Jose & Jini Mol (2024)* highlights the essential requirement for accurate fish species classification to facilitate biodiversity conservation, fisheries management, and environmental monitoring by presenting a deep learning-based methodology for automated identification from underwater photos. The approach employs CNN for feature extraction, capturing intricate visual attributes including color patterns, fin forms, and morphological properties. The study incorporates an improved sequential forward selection (ISFS) technique to augment classification accuracy and model efficiency by picking the most pertinent features. The algorithm, trained on a wide dataset encompassing numerous species and environments, surpasses conventional classification approaches in both accuracy and robustness.

Apart from the Kaggle-based dataset, which is widely recognized for its diverse public datasets, the work in *Pagire & Phadke (2022)* used the Sathyabama Institute of Science and Technology dataset, encompassing 20 different categories of underwater marine organisms with a total of 189 real-time images. The dataset has been pre-processed by implementing image resizing, normalization, noise reduction, and additional color corrections such as color adjustment and contrast enhancement to address specific environmental challenges that are inherent in underwater imaging. The deep learning models applied to this dataset demonstrated exceptional performance, with SqueezeNet leading the best result with an impressive accuracy of 98.41%. DarkNet19 followed closely with an accuracy of 96.30%, while AlexNet achieved a more modest accuracy of 70.45%. These results underscore the effectiveness of deep learning models, particularly SqueezeNet in the classification of underwater organisms.

The success of deep learning methods in image-based classification tasks has increased significantly in recent years. For the purpose of classifying marine creatures, *Tiwari et al. (2023)* employed several deep learning models, including Xception, DenseNet, MobileNet, InceptionV3, VGG16, ResNetV, and ResNet50. Before being used in the training process, these models were first trained on ImageNet where they were refined later using the transfer learning approach. This approach achieved a training accuracy of 95.85% and a validation accuracy of 96.75%, whereby Inception significantly outperformed the other

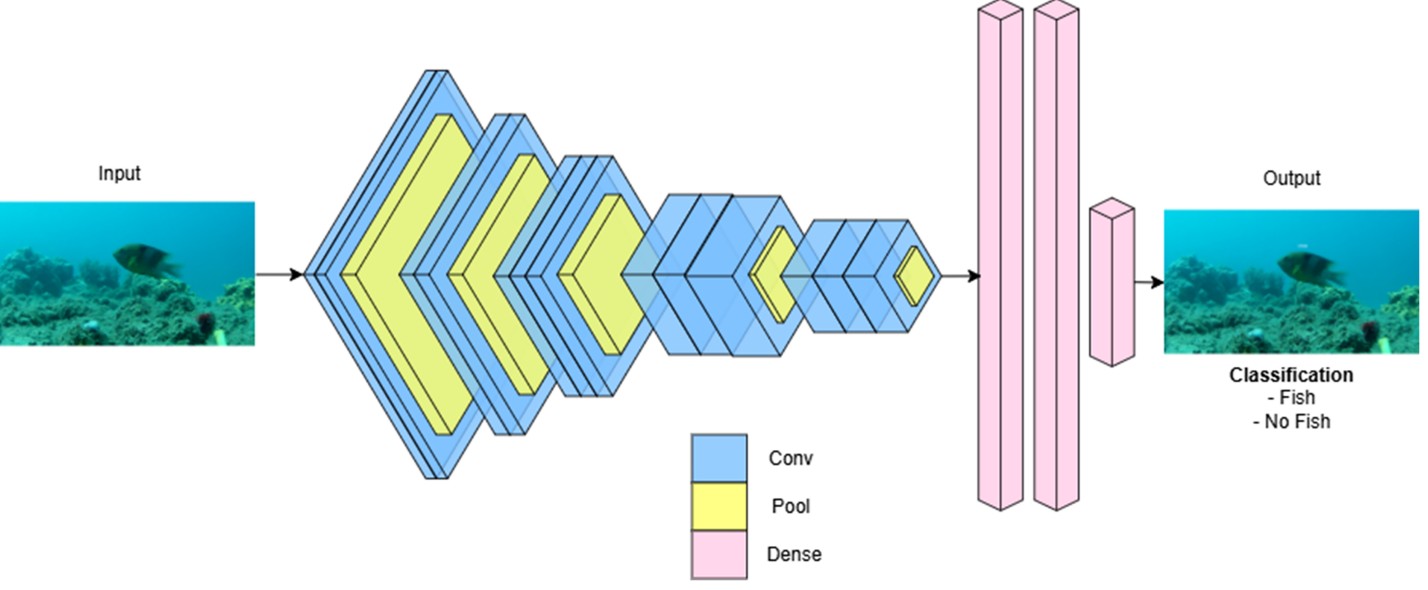

**Figure 3 VGGNet model used for the image classification task.**

tested models. ResNetV also performed well with training and validation accuracies of 93.50% and 93.75%, respectively, while ResNet50 obtained 92.75% and 93.65% training and validation accuracy, respectively. By comparison, Xception performed significantly low with a training accuracy of only 75%, indicating its limitation in categorizing the marine species reliably.

The reviewed CNN models are highly effective for detection and classification tasks, achieving impressive results that set them apart from conventional classification methods. To add on that, in *Shammi et al. (2021)* emphasizes on the classification of six river fish species utilizing CNN model trained on a dataset including 915 images with high resolution. The study highlights the significance of fish classification for consumer knowledge and species comprehension, notably *via* visual characteristics such as the head, body, and scales. Following the downsizing of images to 100 × 100 pixels, preprocessing methods such as normalization and data augmentation increased the dataset to 4,575 images, thereby considerably improving model training. Multiple machine learning algorithms were assessed, with CNN attaining the greatest accuracy of 88.96%, surpassing conventional approaches such as SVM.

All these studies collectively highlight that recent applications of deep learning in aquatic animal husbandry have achieved remarkable automation performance. In summary, CNN-based architectures of deep learning models, such as MobileNet, AlexNet, VGGNet, ResNet, and SqueezeNet, have demonstrated outstanding performance in the classification of aquatic species from underwater images. Figure 3 illustrates the VGGNet model used for the image classification task. Performance is improved through preprocessing techniques, transfer learning, and model-specific modifications, with classification accuracies ranging from 88% to over 99%, according to studies. MobileNet and SqueezeNet are lightweight networks that are well-suited for real-time applications. In

contrast, InceptionV3 and KRS-Net are more complex models that provide enhanced accuracy for datasets with high inter-class similarity. Nevertheless, the generalizability of models across a variety of aquatic datasets continues to be a significant challenge, despite these advancements. Future research should prioritize the evaluation of models that are robust in the face of changing environmental conditions and species diversity. These advancements underscore the growing effectiveness of deep learning techniques in this field. Table 1 provides a detailed overview of the deep learning applications in aquatic animal husbandry.

## LOCALIZATION

For efficient management of the aquaculture industry, localizing the respective aquatic animals within underwater images is just as important as classifying them. Accurately determining the position of these aquatic animals can provide valuable insights into their behavior, population dynamics, and spatial distribution. This kind of data is essential for making well-informed decisions in the aquatic animal husbandry industry, especially when it comes to implementing focused conservation initiatives. Recently, the progress in deep learning development has significantly advanced the ability of the automation algorithms to localize aquatic animals. In this arena, object detection networks such as YOLO and Faster R-CNN have shown encouraging results. These models are capable of identifying and precisely localizing various aquatic species in complex underwater environments (*Litjens et al., 2017*). Research has indicated that deep learning approaches can accurately detect and localize various aquatic species (*Sangari et al., 2023*; *Mathias et al., 2022*) including fish, shrimp, and scallops.

One of the most renowned deep learning models for localization tasks is the YOLO deep network, which provides an efficient and coherent strategy for recognizing fish in challenging underwater images. YOLO is widely known for its speed and accuracy in object detection. Thus, recent studies (*Mathias et al., 2022*; *Wei et al., 2023*; *Rasool, Annamalai & Natarajan, 2024*; *Liu et al., 2023b*; *Raavi, Chandu & SudalaiMuthu, 2023*) have all employed various versions of the YOLO technique.

The faster convergence mode, YOLOv3 was used in *Barui et al. (2018)* to implement underwater object detection. YOLOv3 was also employed in *Mathias et al. (2022)* to locate aquatic species in four challenging video datasets: the University of Western Australia (UWA) dataset, the bubble vision dataset, the DeepFish dataset, and the Life Cross Language Evaluation Forum (CLEF) benchmark from the F4K data repository. The model achieved impressive accuracy rates for fish identification: 97.5% on the CLEF dataset, 96.77% on the UWA dataset, 97.99% on the Bubble Vision dataset, and 95.3% on the DeepFish dataset. In contrast, the same version of YOLO was applied to a custom dataset in *Raavi, Chandu & SudalaiMuthu (2023)*, yielding a precision of 88% and a recall value of 86%, which are notably lower performances compared to those reported in *Mathias et al. (2022)*.

Moreover, a lightweight underwater target detection network was proposed based on a small-scale version of YOLOv5 (YOLOv5s) (*Wei et al., 2023*). The results obtained the network's efficiency, achieving a frame rate (FPS) of 109.12 and a mean average (mAP) at

0.5% of 84.32. Another notable application of YOLO is presented in *Liu et al. (2023a)*, where the algorithm was built upon YOLOv7, which was further enhanced with the backbone of the convolutional block attention module (CBAM). The study utilized rockfish images from the "Label Fishes in the Wild" dataset, published by the National Fisheries Services, and introduced the underwater image enhancement model (UWCNN) for image preprocessing. The enhanced model outperformed the original YOLOv7 model by 3.5%, as demonstrated by the experimental findings, which exhibited a mAP of 94.4% on the testing dataset. Furthermore, the model showed enhanced detection ability in intricate underwater situations with a precision of 99.1% and a recall rate of 99%. The method for detecting fish in an underwater environment suggests that this automated localization task holds significant practical value in promoting the conservation of marine ecosystems and the protection of fish species (*Liu et al., 2023b*).

Furthermore, another localization variant, the region-based convolutional neural network (RCNN) model's primary limitation lies in the need to extract features for each recommended region, which can be computationally intensive. This challenge was mitigated by enhancing the RCNN model through the application of CNN forward computation to the entire image (*Zhang & Yang, 2022*). In *Sangari et al. (2023)*, the detection and classification of marine organisms using RCNN, known for its speed in object detection, was proposed. A correlation filter tracking algorithm (CFTA) was developed to overcome the difficulties associated with tracking and detecting the object of interest in underwater environments, resulting in an impressive overall accuracy of 97.89%. In another study, *Xu et al. (2020)* introduces a fish detection system for aquatic situations utilizing a region-based RCNN, evaluating its efficacy against a Haar feature-based cascade classifier. Monitoring of fisheries is essential for sustainability; yet, conventional techniques employing divers or costly sonar equipment possess inherent limits. A total of 200 images with tags were created for model training using underwater video footage of bluefin tuna. The Haar cascade model, despite its efficiency, encountered difficulties with overlapping fish, resulting in an accuracy of 53.8%. On the other hand, the RCNN model, executed with the PyTorch-based mmdetection toolbox, used selective search for region extraction and attained a much superior accuracy of 92.4%, therefore diminishing false positives and enhancing detection reliability. This study achieved 91.5% detection rate and 92.4% accuracy. Notably, the accuracy in *Xu et al. (2020)* was 5.49% lower than that reported in *Sangari et al. (2023)*.

Besides, *Liu et al. (2023a)* analyses the use of deep learning-based object detection techniques in aquaculture, emphasizing its significance in fish counting, body length assessment, and behavioral analysis to enhance aquaculture management and productivity. It highlights the benefits of non-invasive machine vision systems for acquiring high-quality data, while simultaneously confronting problems such as ambient fluctuation, occlusion, and restricted dataset diversity. The study examines both public datasets (*e.g.*, Fish4Knowledge, LifeCLEF, NOAA) and on-site datasets, addressing image preprocessing techniques such as denoising, enhancement, and augmentation, including the use of GANs for synthetic data production. It classifies object detection techniques into

two-stage models (*e.g.*, R-CNN, Faster R-CNN, Mask R-CNN) and one-stage models (*e.g.*, YOLO), highlighting their respective trade-offs in speed and accuracy.

Then, *Cui et al. (2024)* highlights the rising importance of digital technologies in aquaculture, particularly the application of computer vision, deep learning, and multimodal data fusion for fish monitoring, counting, and behavior analysis. It examines numerous techniques, encompassing 2D and 3D visual tracking, audio tagging, and biosensors, while highlighting obstacles such as occlusions, environmental unpredictability, and data constraints. The research contrasts classical tracking algorithms with deep learning-based methods such as YOLO, StrongSORT, and OC-SORT, while also examining fish counting techniques utilizing infrared, resistivity, acoustic, and computer vision technologies. *Li et al. (2024)* develops an automated fish detection system utilizing infrared cameras for deep-sea aquaculture, with the objective of enhancing operational safety and monitoring efficiency in seawater aquafarming. Images obtained from an infrared camera mounted on a deep-sea net cage were utilized to generate a labeled dataset for training object detection algorithms. The Faster R-CNN framework was utilized, with trials contrasting various backbone networks, enhancement modules, learning rates, and data augmentation techniques. Results showed that using EfficientNetB0 with a feature pyramid network (FPN) provided the best balance of accuracy and speed, achieving an AP50 of 0.85, while the VGG16 configuration reached a slightly higher AP50 of 0.86 at the cost of longer detection time.

Other than YOLO and RCNN approaches, there is also research that suggests using EfficientDet for aquatic animal detection. In *Jain (2024)*, this article's goal was to evaluate how well the newer models perform on the same dataset compared to the previous findings, particularly in terms of accuracy and inference time. This study involved testing and comparing different object detection models on an annotated underwater dataset called the "Brackish Dataset". The projects in this study compared models like YOLOv3, YOLOv4, YOLOv5, YOLOv8, Detectron2 and EfficientDet. The findings showed that the modified EfficentDet outperformed the others, achieving an 88.54% IoU through five-fold cross-validation. Based on these results, the researchers recommended the usage of EfficientDet in complex underwater environments because of its higher accuracy performance.

Apart from that, another localization model, single shot multibox detector (SSD) has also been used in several studies, including *Hu et al. (2020)*, *Li et al. (2022)*, *Raavi, Chandu & SudalaiMuthu (2023)*. In *Hu et al. (2020)*, a feature-enhanced sea urchin detection algorithm was proposed based on the classic SSD algorithm. ResNet-50 was utilized as the network's fundamental architecture to overcome SSD's inability to recognize small targets. The study achieved a mean average precision (mAP) value of 81.0% which is 7.6% greater than the original SSD algorithm, and the confidence rate in detecting small targets has also increased. Similarly, the work in *Li et al. (2022)* improved the model's capability to acquire precise and detailed information about target objects by implementing ResNeXt-50 as the backbone network. This study used two different datasets; one from the National Natural Science Foundation of China underwater Robot Competition and the other from URPC-2019. They reported an accuracy of 79.76%, which is 3.49% higher than the original SSD

algorithm but 1.24% lower than the result produced in *Hu et al. (2020)*. The findings of both investigations were almost identical, indicating that the suggested methods are very effective and suitable for real-time object detection.

Other than that, *Pokharkar (2024)* examines the application of CNN for automatic detection of fish species in underwater environments, aiming to improve accuracy and efficiency compared to conventional traditional methods. The study overcomes major challenges, including inadequate visibility from environmental influences, fish concealment, and diminished image quality resulting from motion or resolution constraints, by training CNNs on extensive datasets. The suggested model exhibits robust efficacy in differentiating diverse fish species, providing essential resources for marine scientists and conservationists to assess biodiversity and ecosystem vitality. *Li et al. (2023)* examines how deep learning provides potential tools for intelligent marine exploration, despite its challenging application in aquatic environments due to visibility constraints, animal diversity, and complex underwater circumstances. The article examines contemporary applications of deep learning, including image and video identification, species classification, biomass calculation, behavior analysis, and seafood safety monitoring.

As mentioned above, YOLO and Faster R-CNN have shown promising results in classifying and localizing various aquatic species, particularly in complex underwater environments. In *Raavi, Chandu & SudalaiMuthu (2023)*, both of these core deep learning frameworks, YOLOv8 and Faster R-CNN were used to localize the objects of interest, leading to significant improvements in the efficiency and accuracy of marine life detection. The study reported an mAP of 96.1 for YOLOv8 and 96.4 for Faster R-CNN models. Additionally, the study utilized two different data sources, a GitHub repository and Google Pictures, which further demonstrates the robustness of these models in detecting and identifying aquatic species accurately, highlighting their effectiveness in challenging underwater scenarios.

In summary, deep learning-based localization methods have demonstrated strong effectiveness in identifying and tracking aquatic animals within underwater environments. These recent studies have also shown that various versions of YOLO managed to successfully achieve high localization accuracy. The real-time performance and accuracy of YOLO variants, particularly YOLOv3, YOLOv5s, YOLOv7 with CBAM, and YOLOv8, are consistently high. YOLOv5s has a remarkable speed of 109.12 FPS, while YOLOv7+CBAM achieves mAP values exceeding 94%. Similarly, Faster R-CNN models, particularly those that are incorporated with EfficientNetB0 and FPN, have demonstrated robust detection performance (AP50 as high as 0.86), although at the cost of extended inference times. Figure 4 illustrates the working process of the YOLO object detection model applied to aquatic environments. Other architectures such as SSD, EfficientDet, and RCNN have also been applied, with enhancements like ResNet-50 and ResNeXt-50 improving detection of small or occluded targets. The consistent deployment of underwater imaging technologies in real-world aquaculture environments continues to be a significant challenge due to the variability in underwater imaging conditions, dataset limitations, and computational trade-offs, despite these advancements. This underscores the model's effectiveness across

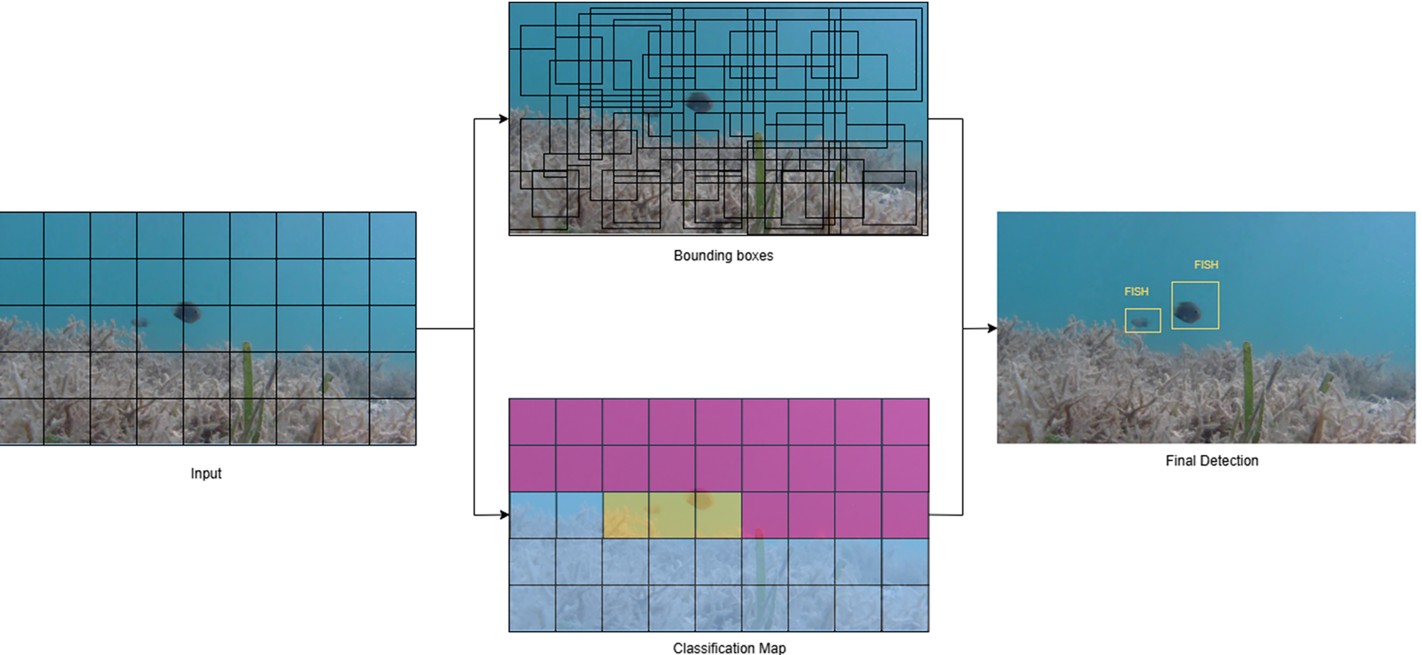

**Figure 4 Working process of the YOLO object detection model applied to aquatic environments.**

different scenarios in aquatic animal husbandry. Table 2 shows a detailed overview of the deep learning-based localization models in aquaculture farms.

## SEGMENTATION

Segmentation is another critical automation task in the aquaculture industry, as it enables precise identification of individual aquatic animals within an image. The goal of semantic segmentation is to provide a label for every pixel in an image by breaking it up into segments based on its semantic content. In general, it refers to segmenting all the pixels into different object categories, attempting to split the picture into semantically relevant pieces, and categorizing each portion into one of the predetermined classes using semantic segmentation techniques (*Elizar et al., 2022*). This process involves classifying each pixel into specific categories according to its characteristics, producing a segmentation map with essential semantic information. Accurate segmentation is key to tasks such as population monitoring, growth analysis, and the automation of feeding systems.

Deep learning-based models that take advantage of CNNs' hierarchical feature extraction capabilities like U-Net have demonstrated exceptional performance in segmenting aquatic animals in complex underwater environments. These models effectively trace the boundaries of individual aquatic organisms, ensuring accurate delineation even in challenging conditions. As *Muñoz-Benavent et al. (2022)* introduces a deep learning approach for the automatic detection, segmentation, and sizing of fish in aquaculture, specifically targeting Bluefin tuna. This research assesses various deep learning models, including Faster R-CNN, YOLOv5, Mask R-CNN, and PointRend, for the segmentation of fish in underwater settings where visibility and motion present

considerable difficulties. PointRend attained superior segmentation accuracy, facilitating enhanced scaling *via* edge detection and model fitting. The application of tracking algorithms and repeated measurements diminished size inaccuracies, illustrating that deep learning technologies surpass traditional techniques in clarity and adaptability. Other research from *Muntaner-Gonzalez et al. (2025)* presents a technique based on deep learning for the detection and segmentation of Mediterranean fish species, addressing the challenges posed by traditional manual biodiversity assessments that are labor-intensive and problematic in intricate underwater environments. Utilizing advancements in computer vision and deep learning, the authors created an open dataset and trained a YOLOv8l-seg model, refined *via* hyperparameter optimization and bespoke data augmentation methods. The model, trained on photos from several sources including the MINKA database, exhibited enhanced accuracy in fish detection and segmentation, especially with higher image sizes and a generic fish class. The model, included into a real-time Stereo Vision System (SVS), successfully detected fish during field trials, although its performance was somewhat diminished by the presence of small fish and cluttered backdrops. In order to achieve accurate segmentation of the underwater Fish4Knowledge image dataset, the study in *Nezla, Haridas & Supriya (2021)* extended and modified a U-Net-based semantic segmentation network. By training and fine-tuning the U-Net model with an optimal set of hyperparameters, the average IoU score reached 0.8583 on the Acanthurus Nigrofuscus class from the Fish4Knowledge dataset. In comparison, the method proposed in *Zhang, Li & Seet (2024)*, which used the same dataset but focused on different classes, achieved a higher mIoU of 94.44%, an mAP of 97.03%, and a frame rate of 43.62 FPS. Their model also demonstrates superior segmentation performance while balancing both accuracy and speed effectively for underwater image analysis. A closer look at both studies shows that while the proposed model in *Nezla, Haridas & Supriya (2021)* offers competitive IoU scores, the results from *Zhang, Li & Seet (2024)* are notably stronger, highlighting improvements in precision and processing efficiency.

In another development, the work in *Zhang, Gruen & Li (2022)* proposed an improved boundary-oriented U-Net model to automatically identify and segment coral individually from orthophotos. The Pocillopora corals can be reliably distinguished, whether they are alive or dead based on the optimized network, contributing to assessments of coral health. This improved U-Net architecture focuses on boundary details and has shown superior performance in segmenting irregular coral edges, which are often challenging to process. The study achieved a small improvement in mIoU, about 0.4%, compared to the original U-Net. The original U-Net had an mIoU of 93.2%, while the enhanced version achieved 93.6% mIoU.

Building upon the U-Net architecture, various convolutional layers and residual blocks have been incorporated for enhanced target detection and segmentation, particularly in situations where training images are limited. The U-Net model has been modified to integrate additional convolutional layers, residual blocks, and SE blocks, with the introduction of a new novel block to emphasize relevant features. This proposed approach was analyzed and compared to the current CNN models using open-sea underwater fish

datasets, demonstrating its efficacy in scenarios with limited training data. The model achieved a performance of 95.04% in F1-score and 88.19% in mIoU in complex scenes. These results highlight its ability to outperform other improved U-Net approaches, including those in *Zhang, Gruen & Li (2022)*. Then, *Zhu, Liao & Xu (2022)* introduces an innovative method to augment underwater target monitoring in the aquaculture sector by refining fish segmentation in low-contrast, color-biased environments. The suggested two-step approach integrates Auto-MSRCR image enhancement, which enhances visibility and contrast in underwater photos, with the U2-Net semantic segmentation algorithm, recognized for its accuracy in object delineation. Experimental findings indicate that the integration of picture enhancement improves segmentation accuracy by roughly 5% relative to models lacking preprocessing, underscoring the significance of visual quality in deep learning efficacy. In *Nguyen et al. (2024)* presents AquaAttSeg, an innovative deep learning model aimed at automating fish size measurement in aquatic environments, mitigating the drawbacks and hazards associated with conventional manual techniques. AquaAttSeg, constructed on a U-shaped architecture, utilizes the PVTv2-B3 encoder for superior semantic information extraction and integrates the HFIM module, substituting conventional skip connections with SAM and CAM modules to boost information flow. A custom Convmix Transpose block in the decoder further augments performance by reinstating lost features during segmentation. AquaAttSeg, assessed using the Deepfish and SUIM datasets, surpassed numerous leading models, exhibiting exceptional accuracy and efficiency in fish segmentation. Then, *Kim & Park (2022)* introduces PSS-net, an innovative deep learning architecture aimed at enhancing marine scene segmentation by tackling significant problems, including low-light underwater environments and the concealment of marine life. PSS-net utilizes a parallel architecture featuring two independent models and loss functions to distinctly process marine life and their habitats, hence improving segmentation accuracy, in contrast to earlier models that segment foreground and background concurrently. The incorporation of an attention mechanism enhances the model's capacity to differentiate between intricate visual components. The final segmentation is accomplished by integrating feature maps from both branches, yielding a highly precise depiction of underwater sceneries. PSS-net demonstrated exceptional performance on a public marine animal segmentation dataset, achieving 87% mIoU, 97.3% structural similarity, and 95.2% enhanced alignment, surpassing previous methodologies.

In *Liu & Fang (2020)* presented a semantic segmentation network for underwater images based on improved versions of the DeepLabv3+ network. The dataset used for this study consists of a combination of self-made underwater images, some of which were sources from open resources on the Internet, while others were captured by a laboratory underwater robot (HUBOS-2K, Hokkaido University). The study builds upon the current DeepLabv3+ framework, optimizing and fine-tuning relevant parameters to enhance its performance. The findings indicate that the suggested approach significantly improves the segmentation of target objects, particularly in preserving the appearance and boundaries of the segmented targets, while preventing pixel mingling between classes. This method

increases segmentation accuracy by 3% compared to the original approach, achieving an mIoU of 64.65% on the underwater dataset.

The semantic segmentation of underwater imagery (SUIM) dataset is the first large-scale dataset specifically designed for underwater object segmentation. The SUIM dataset, which comprises more than 1,525 images with pixel-level annotations, comprises eight object categories (fish, reefs, aquatic plants, wrecks/ruins, human divers, robots, and the sea floor), was used in *Islam et al. (2020)* to assess the performance of various state-of-the-art (SOTA) semantic segmentation models. This study introduces SUIM-Net, a fully convolutional encoder-decoder model that offers a significantly faster runtime compared to SOTA methods while maintaining competitive segmentation accuracy. The model achieves a parameter count of 3.864 million, a frame rate of 28.65 fps, a mIoU of 77.77 ± 1.64, and an F1-score of 78.86 ± 1.79.

In *Chicchon et al. (2023)*, the same SUIM dataset, along with two other publicly accessible datasets, RockFish and DeepFish, were employed to evaluate segmentation models for underwater imagery. These datasets include pixel annotations of three class categories: background water, seafloor/obstacles, and fish. The study tests CNN-based architectures, namely U-Net and DeepLabv3+, using different loss functions, which are cross-entropy, dice, and active contours. U-Net with attention mechanisms (scSE blocks) in the decoder layers outperformed the more complex DeepLabv3+ architecture. When combined with the lightweight EfficientNet-B0, U-Net-scSE achieved an mIoU of 86% and an mHD95 of 26.61 mm. When paired with heavier architectures like EfficientNet-B7 and ResNeSt-296e, the model's mIoU improved to 87.45%, with an mHD95 of 23.40 mm. This highlights the significant advantage of incorporating attention mechanisms and advanced backbone architectures to boost segmentation performance in underwater environments.

Another relevant approach is proposed in *Drews-Jr et al. (2021)*, which developed a dataset of real underwater images combined with simulated data to train two of the top-performing deep learning segmentation architectures. The goal was to tackle the challenge of segmenting underwater images in the wild. This study introduced a dataset specifically designed to train deep CNN architecture for underwater image segmentation. The research demonstrated a working solution using the DeepLabv3+ and SegNet architectures, achieving an mIoU of 0.919 and 0.825, respectively, on a random test set of 300 real underwater images. Notably, these architectures were able to perform accurate segmentation with only a small number of training images, showing their effectiveness in underwater environments where labeled data is often limited.

The SegNet model discussed in *Drews-Jr et al. (2021)* achieved slightly lower results compared to *O'Byrne et al. (2018)*, where an mIoU of 87% and a mean accuracy of 94% were reported. The difference in performance between *Drews-Jr et al. (2021)*, *O'Byrne et al. (2018)* is about 4.5%. A total of 2,500 annotated synthetic images with a resolution of 960 × 540 pixels, were used to train SegNet in *O'Byrne et al. (2018)*. After training on the synthetic data, SegNet was successfully applied to segment new real-world underwater images.

There are various segmentation architectures designed for aquatic animal husbandry. In *Han et al. (2023)*, a method based on an improved PSPNet network (IST-PSPNet) was

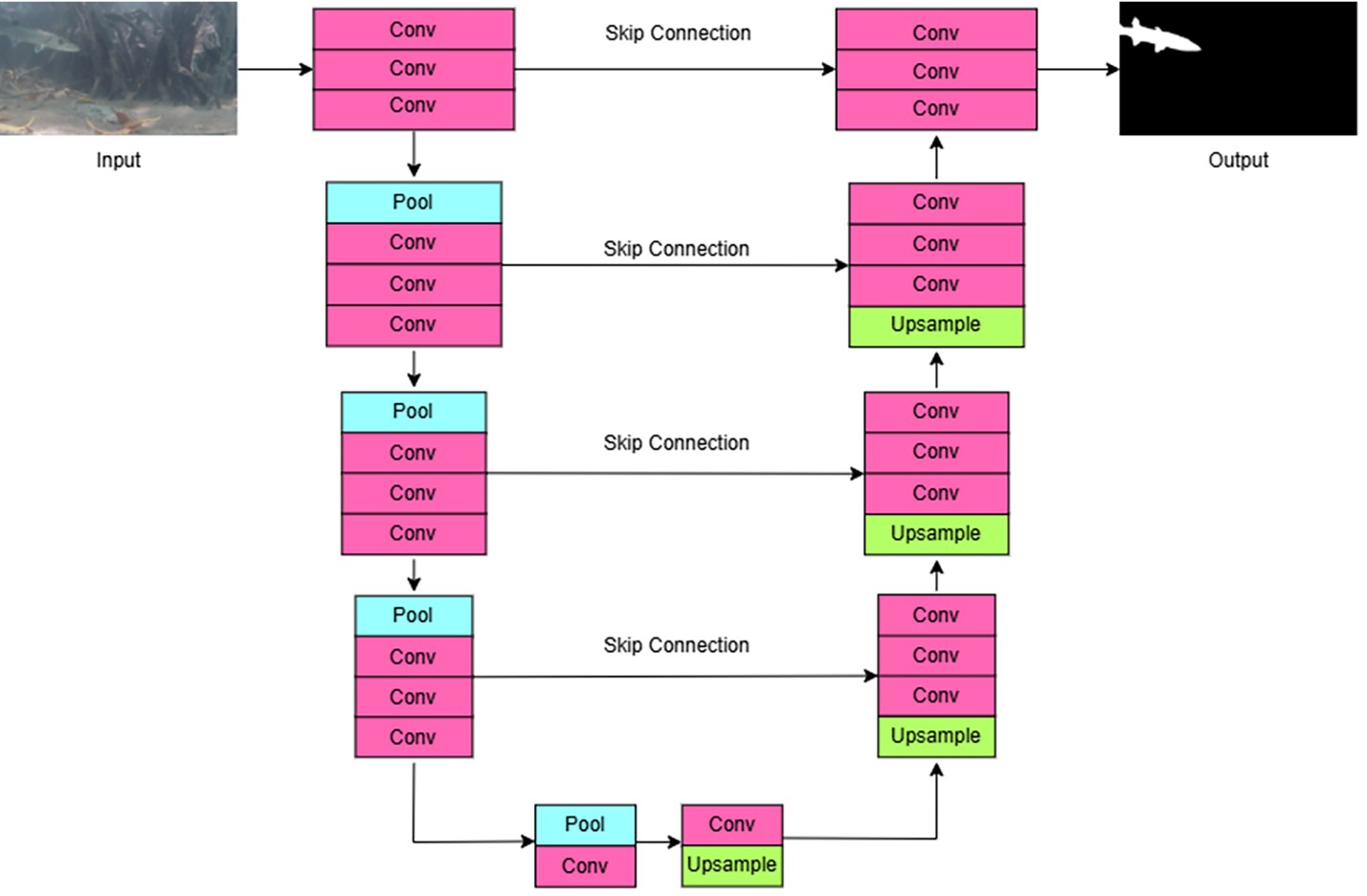

**Figure 5  U-Net architecture for the semantic segmentation task.**

proposed for underwater fish segmentation. The results demonstrated superior performance on the DeepFish underwater image dataset, achieving a 91.56% mIoU, 46.68M parameters, and 40.27 giga floating-point operations per second (GFLOPS). This method showed significant improvements in accurately segmenting fish with similar colors and challenging backgrounds, especially for smaller fish. These enhancements highlight its effectiveness in overcoming the typical challenges of underwater image segmentation.

In summary, segmentation models such as U-Net, DeepLabv3+, and SUIM-Net showed high accuracy in identifying aquatic organisms in various underwater environments. Improvements such as attention methods, residual blocks, and innovative encoder-decoder architectures (*e.g.*, HFIM in AquaAttSeg, dual-path structures in PSS-net) have significantly augmented segmentation accuracy, especially in low-light, complex, or color-biased environments. Lightweight U-Net variants with efficient backbones, such as EfficientNet-B0, optimize speed and accuracy, while methods like image augmentation (*e.g.*, Auto-MSRCR) and domain-specific preprocessing enhance model robustness. Figure 5 shows the U-Net architecture for the semantic segmentation task. Despite top-performing models such as AquaAttSeg, PSS-net, and enhanced PSPNet attaining

mIoU scores exceeding 90%, issues remain in managing small or occluded objects, real-time processing, and generalization across datasets, necessitating additional optimization. Table 3 shows a detailed overview of the applications of deep learning-based segmentation models in aquaculture monitoring systems. This detailed comparison allows for a better understanding of the advancements in segmentation models and their effectiveness in addressing the complexities of underwater environments.

## GENERAL DISCUSSION

While deep learning models have undeniably advanced the classification, localization, and segmentation tasks in aquatic animal husbandry, several challenges and limitations need to be addressed before these applications can be broadly applied in real-world settings. These limitations affect the core tasks essential for monitoring, health assessment, and automation in underwater environments. Tables 1–3 summarizes the model performance, dataset characteristics, strength and weakness and of different deep learning models applied in previous studies. In evaluating the performance of deep learning models for underwater object detection, computational cost and time complexity play a crucial role. However, not all studies explicitly report these metrics, as they are highly dependent on factors such as hardware specifications, dataset size, and implementation details. This structured analysis allows us to assess the strengths and limitations of different methodologies, addressing model suitability for real-world aquaculture applications. For example, CNN-based models such as U-Net and DeepLabv3+ demonstrate strong performance in segmentation tasks but often require large annotated datasets for effective training. On the other hand, models like YOLO and Faster R-CNN prioritize real-time efficiency but may face trade-offs in accuracy depending on environmental conditions. These trade-offs must be carefully considered based on specific application requirements in detecting underwater image. Future research should explore lightweight architectures and model compression techniques to improve deployment feasibility in real-world aquaculture settings.

### Limited availability of quality datasets

A major obstacle across all tasks (classification, localization, or segmentation) is the lack of large, annotated datasets specific to underwater environments. Often we find that existing studies rely on synthetic or small-scale manually annotated datasets, which fail to capture the full range of real-world underwater conditions. From a technical perspective, deep learning models require large, annotated datasets, but underwater environments introduce challenges such as poor visibility, varying lighting conditions, and water turbidity. Most available datasets are small-scale or synthetic, making it difficult for models to generalize to real-world conditions. As a result, the models tend to perform well in controlled setups but may not generalize when exposed to diverse species, habitats, or environmental conditions. For instance, while segmentation models such as U-Net and DeepLabv3+ perform well on specific datasets, they may struggle when applied to other underwater ecosystems or animal species. Expanding datasets to include a broader variety of species, environments, and lighting conditions would significantly enhance model robustness across these aquatic

tasks. Then, annotating underwater images is a labour-intensive process that necessitates expert knowledge, which complicates the process of obtaining large labelled datasets from a practical perspective. Annotating underwater images is a labour-intensive process that necessitates expert knowledge, which complicates the process of obtaining large labelled datasets from a practical perspective. The necessity of collaborative data-sharing platforms to enhance the diversity and accessibility of datasets is underscored by the fact that numerous aquaculture facilities and marine research institutions lack the infrastructure necessary for systematic data collection. Lastly, the behaviour and habitats of marine life may be impacted by the frequent human intervention in underwater data collection, which may disrupt aquatic ecosystems from an environmental perspective. In order to mitigate ecological damage and guarantee a consistent supply of high-quality data for deep learning applications, it is imperative to prioritise passive monitoring systems and AI-driven autonomous data acquisition methods. Besides dataset limitations, underwater images often suffer from blurriness, low contrast, and color distortions due to light scattering and absorption, affecting the accuracy of deep learning models. To address these issues, preprocessing techniques such as contrast enhancement, color correction, and denoising filters are commonly applied before training. Recent methodologies have investigated deep learning augmentation techniques, including GAN-based restoration and convolutional autoencoders, that improve image clarity. Confronting these issues with a multi-faceted strategy will be essential for progressing deep learning in underwater object detection.

## Accuracy *vs.* speed in real-time applications

Balancing between accuracy and computational speed remains a critical challenge for real-time applications, especially for localization or object detection tasks. From a technical perspective, many models either excel in accuracy or computational speed, but struggle to perform well in both metrics. For example, while Faster R-CNN and YOLO show impressive results in detecting objects, they either sacrifice speed for accuracy or vice versa, making them difficult to deploy in real-time monitoring scenarios, like fish farms or natural habitats. To improve real-time processing, future research should focus on lightweight architectures, model quantization, and pruning techniques. In the practical perspective, real-time detection is crucial for applications like fish counting, behavior monitoring, and disease detection, yet many deep learning models struggle with the high computational demands required for real-time performance. Developing models that achieve an optimal balance between real-time performance and precision is a key research direction, especially as more commercial applications emerge for monitoring fish populations. Lastly, from an environmental perspective, the implementation of AI-driven monitoring systems must take into consideration sustainability and energy usage. High-performance computing solutions lead to heightened energy consumption and carbon emissions, necessitating future study to investigate sustainable AI options, such as enhancing model efficiency and incorporating renewable energy sources into underwater monitoring stations. Comprehensively addressing these perspective will be essential for assuring the feasibility, affordability, and sustainability of deep learning applications in underwater object detection.
## Generalization across species

A frequently overlooked challenge is the need for models to generalize across different species and ecosystems. From a technical perspective, transfer learning, meta-learning, and domain adaptation techniques could help models become more robust across diverse aquatic ecosystems. The practical perspective lies in the significant variation among fish species in terms of size, shape, color, and behavior, making it difficult for models to generalize. Many of the existing models, while performing well on a specific set of data, may not be able to adapt to new environments or species without significant retraining. This presents a challenge for scaling these applications to large, diverse aquatic ecosystems. Future work could focus on developing models that are flexible enough to handle varying species without requiring extensive retraining and fine-tuning. Finally, from an environmental perspective, misclassifications in automated monitoring systems can lead to inaccurate assessments of fish populations, potentially influencing conservation efforts and fisheries management. Despite these challenges, as deep learning architectures continue to evolve and datasets become more comprehensive, these applications will likely transition from experimental phases to practical applications in aquatic animal husbandry. Collaboration between researchers and industry practitioners will also be essential in ensuring that these models are designed with real-world constraints in mind.

## CONCLUSION AND FUTURE WORKS

This review article highlights how deep learning has significantly transformed the field of aquatic animal husbandry, particularly in tasks like classification, localization, and segmentation. Models such as U-Net, with its remarkable segmentation accuracy of 94.44%, clearly outperform conventional methods, showcasing the potential of deep learning to address real-world challenges. While existing research has made significant strides in classification, localization, and segmentation tasks for underwater object detection, several critical challenges remain. Beyond the well-known issues of dataset diversity and image quality, practical deployment faces hurdles such as real-time processing constraints, model robustness across diverse aquatic environments, and the need for adaptive learning techniques. Overcoming these issues will be essential to unlocking the full potential of deep learning in this field.

Looking ahead, future efforts should focus on addressing these challenges. Expanding dataset diversity and improving the adaptability of models across different environments and species will be crucial steps. Enhancing image quality, particularly in poor lighting conditions or turbid water, will also play a vital role in strengthening model robustness. Additionally, exploring advanced architectures, such as those incorporating attention mechanisms, could further improve segmentation accuracy, especially in more complex scenarios. Futhermore, self-supervised and domain adaption techniques could enhance model generalisation without the necessity for significant labelled data, tackling the issue of diverse environmental conditions. Moreover, multi-modal fusion techniques that combine sonar, LiDAR, or acoustic data with optical inputs should improve object detection efficacy in difficult underwater conditions. With these advancements, deep learning can continue to revolutionize automated monitoring systems for aquatic animal husbandry.

### Funding

This project is supported by the Ministry of Higher Education Malaysia through Fundamental Research Grant Scheme (FRGS/1/2022/TK07/UKM/02/4) and the Universiti Kebangsaan Malaysia through Dana Padanan Kolaborasi (DPK-2023-006). The funders had no role in study design, data collection and analysis, decision to publish, or preparation of the manuscript.

### Grant Disclosures

The following grant information was disclosed by the authors:
Ministry of Higher Education Malaysia through Fundamental Research: FRGS/1/2022/TK07/UKM/02/4.
Universiti Kebangsaan Malaysia through Dana Padanan Kolaborasi: DPK-2023-006.

### Competing Interests

The authors declare that they have no competing interests.

### Author Contributions

- Marzuraikah Mohd Stofa analyzed the data, performed the computation work, prepared figures and/or tables, authored or reviewed drafts of the article, and approved the final draft.
- Fatimah Az Zahra Azizan conceived and designed the experiments, performed the experiments, analyzed the data, performed the computation work, prepared figures and/or tables, authored or reviewed drafts of the article, and approved the final draft.
- Mohd Asyraf Zulkifley conceived and designed the experiments, performed the experiments, analyzed the data, performed the computation work, prepared figures and/or tables, authored or reviewed drafts of the article, and approved the final draft.

### Data Availability

This is a review article and did not generate raw data.

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
