# Peer review of "A review of deep learning methods in aquatic animal husbandry"

_PeerJ Computer Science, doi:10.7717/peerj-cs.3105_

## Round 0.1 · original submission · Major Revisions

Dear authors,

Thank you for submitting your article. Feedback from the reviewers is now available. It is not recommended that your article be published in its current format. However, we strongly recommend that you address the issues raised by the reviewers, especially those related to readability, experimental design and validity, and resubmit your paper after making the necessary changes.

Best wishes,

·

Basic reporting

The submitted articles are cross-disciplinary in nature, namely aquaculture and computer science (in this case artificial intelligence using deep learning), and fall within the scope of the Peer J. In recent times, the topic of the use of AI in the cultivation of aquatic organisms has attracted quite a lot of attention as it related to food security.

The field proposed has been studied recently. However, there is still room to look at it in other aspects, not only what methods have been used as stated in this article but also examine critically the whole process from the data acquisition to visualization of the results where deep learning can contribute at most.

The subject discussed has not been introduced adequately, is still too general and not clear enough about who is being targeted, whether algorithm developers or users/practitioners of artificial intelligence.

Experimental design

The search method is quite complete and consistent, but the scope still does not include critical subjects and underwater object studies, namely preprocessing or image enhancement, considering that underwater photos or images are always blurry so that there needs to be primary attention. In addition, objects in this case organisms in water are dynamic and always moving so that tracking issues need to taken into consideration.

Sources have been cited based on scope or keywords and adequately quoted and paraphrased.

The literature review conducted have not been well organized and logical, and there is no clear development over time of how the methods cited contribute to the development of knowledge, resulting in improvements or refinements of the learning algorithm.

Validity of the findings

This literature review article attempts to present the latest developments in detecting underwater objects using deep learning methods. In addition, it is also stated that a comparative analysis of the performance of the various methodologies presented in this literature review article will be carried out. However, the review of this article has not systematically described the contribution of each model used that model makes to improve underwater detection performance. It is suggested to insert additional column in the Table1 1-3 that provide information on the specific feature of the model that is/are distinct from other similar model.

The conclusions given in this literature review article states that the major challenges faced are the limited diversity of datasets and the quality of underwater images. The direction of future research is mentioned on how to answer these challenges. Actually, what is conveyed here is not something new and should have been known and realized since starting of this review. Unfortunately, this aspect has received less consideration, while the issues of classification, localization and segmentation that are the focus of this review are actually relatively well-established and no longer posed a major difficulty in identifying direction of future research.

Additional comments

The literature review article submitted does not yet provide new perspectives related to the challenges and direction of future research in underwater object detection, especially in the cultivation of aquatic organisms (aquaculture). What are the main challenges have not been conveyed sharply based on the latest developments in knowledge, both in terms of underwater images and deep learning algorithms.

Cite this review as

Reviewer 2 ·

Basic reporting

This paper reviewed deep learning methods in aquatic animal husbandry.

Experimental design

The authors only focused on classification, localization and segmentation tasks of deep learning methods in aquatic animal husbandry.
There are some concerns the authors should take care of as follows:
-The authors should give a research question.
- They should describe their collected papers in detail.
- Why did they only focus on classification, localization and segmentation tasks?
- They should present all deep learning techniques in detail.
- Their experiments are not strong.
- They should discuss in dimensions.

Validity of the findings

- They didn’t compare with any research, they should compare with a review paper in the same subject.
- They should do more experiments and analyze the result in detail.
- They should collect more advanced methods in recent year.

Cite this review as

·

Basic reporting

The presented scientific review is devoted to the application of deep learning methods in automated systems for monitoring aquatic animals. In the context of growing demand for seafood and the need to improve the efficiency of aquaculture, the use of artificial intelligence technologies is becoming increasingly important. The authors consider key approaches to image and video processing, paying special attention to the problems of classification, localization and segmentation of aquatic organisms. This review is relevant and useful, since it systematizes modern research and gives an idea of ​​the prospects for using deep learning in this area.
The authors consider three key tasks of computer vision in the context of aquaculture:
1. Classification - automatic identification of aquatic animal species based on images.
2. Localization - determining the exact location of objects in the image.
3. Segmentation - detailed selection of object boundaries for more accurate analysis.
The review analyzes in detail modern neural network architectures, such as CNN, YOLO, U-Net, as well as their performance in solving various problems. Examples of studies demonstrating high accuracy of models are presented, for example, segmentation algorithms based on U-Net achieve 94.44% accuracy. The authors also highlight the existing challenges, including the limited datasets, the impact of underwater conditions on image quality, and the need to improve data processing methods.
The article is written in clear, unambiguous English using technically correct wording. As a non-native speaker, reading the work did not cause any difficulties for understanding
The article includes a sufficient introduction, providing the context of the field and explaining its relevance. The authors analyze publications from authoritative databases (IEEE Xplore, Google Scholar, ScienceDirect), which confirms the high level of reliability of the materials.
The structure of the article is logical, the design and presentation of the work do not cause any comments.

Experimental design

The article corresponds to the stated objectives. The review methodology with the search strategy seem reasonable. The selected classes of solved problems and solution options seem sufficient.
As a wish, it is possible to recommend expanding the comparison of different approaches with an indication of their strengths and weaknesses in specific application conditions. The problem of generalizing models for work in real conditions has not been sufficiently considered: it is difficult to assess how effective they will be under changing environmental parameters.
In addition, within the comparison in the table, it may be appropriate to mention the consumption of computing and time resources, at least one line for each architecture in the table articles.

Validity of the findings

The conclusions are logically linked to the main objective of the review and are substantiated; the argumentation is built consistently.

Additional comments

Overall, "A review of deep learning methods in aquatic animal husbandry" is a valuable scientific work that provides up-to-date information on the application of deep learning methods in the field of aquaculture. The authors successfully summarize existing research and suggest possible ways to develop it. Despite some shortcomings, such as limited analysis of the shortcomings of the models and their applicability in real-world conditions, the work makes a significant contribution to the development of this scientific field.

Cite this review as

---

## Round 0.2 · Minor Revisions

Dear Authors,

One of the previous reviewers did not respond to the invitation for reviewing the revised paper. Although one reviewer thinks that your paper can be accepted, one reviewer suggests major revision. We encourage you to address the concerns and criticisms of Reviewer 2 and resubmit your paper once you have updated it accordingly.

Best wishes,

Reviewer 2 ·

Basic reporting

In research question, they should number their questions and answer each question in detail on the content of the paper.
In the revision, they didn’t present their collected papers in detail. Because of this importance in the reviewing, they should present why, where and how to collect these papers in detail.
They should present the main point of each deep learning technique on the paper.

Experimental design

They should compare with some review papers in the same subject.
They should collect more papers in recent year, do more experiments and analyze the result in detail.
I keep these comments on this review.

Validity of the findings

Overall, I saw that the revision was not changed much, they should major revise the paper.

Cite this review as

·

Basic reporting

The presented scientific review is devoted to the application of deep learning methods in automated systems for monitoring aquatic animals. In the context of growing demand for seafood and the need to improve the efficiency of aquaculture, the use of artificial intelligence technologies is becoming increasingly important. The authors consider key approaches to image and video processing, paying special attention to the problems of classification, localization and segmentation of aquatic organisms. This review is relevant and useful, since it systematizes modern research and gives an idea of ​​the prospects for using deep learning in this area.
The authors consider three key tasks of computer vision in the context of aquaculture:
1. Classification - automatic identification of aquatic animal species based on images.
2. Localization - determining the exact location of objects in the image.
3. Segmentation - detailed selection of object boundaries for more accurate analysis.
The review analyzes in detail modern neural network architectures, such as CNN, YOLO, U-Net, as well as their performance in solving various problems. Examples of studies demonstrating high accuracy of models are presented, for example, segmentation algorithms based on U-Net achieve 94.44% accuracy. The authors also highlight the existing challenges, including the limited datasets, the impact of underwater conditions on image quality, and the need to improve data processing methods.
The article is written in clear, unambiguous English using technically correct wording. As a non-native speaker, reading the work did not cause any difficulties for understanding
The article includes a sufficient introduction, providing the context of the field and explaining its relevance. The authors analyze publications from authoritative databases (IEEE Xplore, Google Scholar, ScienceDirect), which confirms the high level of reliability of the materials.
The structure of the article is logical, the design and presentation of the work do not cause any comments.

Experimental design

The article corresponds to the stated objectives. The review methodology with the search strategy seem reasonable. The selected classes of solved problems and solution options seem sufficient.
The comments made during the previous iteration of the review have been taken into account, the corresponding edits have been made to the article, there are no new comments, the work can be accepted for publication.

Validity of the findings

The conclusions are logically linked to the main objective of the review and are substantiated; the argumentation is built consistently.

Additional comments

Overall, "A review of deep learning methods in aquatic animal husbandry" is a valuable scientific work that provides up-to-date information on the application of deep learning methods in the field of aquaculture. The authors successfully summarize existing research and suggest possible ways to develop it.
The comments made during the previous iteration of the review have been taken into account, the corresponding edits have been made to the article, there are no new comments, the work can be accepted for publication.

Cite this review as

---

## Round 0.3 · Minor Revisions

Dear Authors,

We encourage you to address the minor concerns and criticisms of Reviewer 2 and resubmit your paper once you have updated it accordingly.

Best wishes,

Reviewer 2 ·

Basic reporting

The revision is better.

Experimental design

The figure 1 should be revised.

Validity of the findings

The authors should use more figures to visualize the result of the paper.
because it has only Figure 1.
Here are some suggestions:
- Figure describes collected papers
- Figure draws technique
- Figure presents architecture of some main deep learning techniques
- Figure presents application of this topic ...

Cite this review as

---

## Round 0.4 · accepted · Accept

Dear Authors,

Thank you for addressing the reviewers' comments. Your manuscript now seems sufficiently improved and ready for publication.

Best wishes,

Reviewer 2 ·

Basic reporting

The revised paper includes my comments.

Experimental design

I accept the revision

Validity of the findings

I don't have comment more.

Cite this review as